# Dietary Acid Load Correlates with Serum Amino Acid Concentrations after a Four-Week Intervention with Vegan vs. Meat-Rich Diets: A Secondary Data Analysis

**DOI:** 10.3390/nu15132942

**Published:** 2023-06-28

**Authors:** Julian Herter, Ann-Kathrin Lederer, Alvaro Luis Ronco, Luciana Hannibal, Roman Huber, Maximilian Andreas Storz

**Affiliations:** 1Department of Internal Medicine II, Centre for Complementary Medicine, Freiburg University Hospital, Faculty of Medicine, University of Freiburg, 79106 Freiburg, Germanyann-kathrin.lederer@uniklinik-freiburg.de (A.-K.L.); roman.huber@uniklinik-freiburg.de (R.H.); 2Department of General, Visceral and Transplant Surgery, University Medical Center of the Johannes Gutenberg University, 55131 Mainz, Germany; 3Unit of Oncology and Radiotherapy, Pereira Rossell Women’s Hospital, Bvard. Artigas 1590, Montevideo 11600, Uruguay; 4Laboratory of Clinical Biochemistry and Metabolism, Department of General Pediatrics, Adolescent Medicine and Neonatology, Faculty of Medicine, Medical Center, University of Freiburg, 79106 Freiburg, Germany; luciana.hannibal@uniklinik-freiburg.de

**Keywords:** potential renal acid load, dietary acid load, vegan, plant-based diet, amino acids, protein, glutamine, glycine, lysine

## Abstract

Chronic low-grade metabolic acidosis is now a common phenomenon in the Western world. The high dietary intake of sulfur-containing amino acids in the form of processed meats results in an excessive release of acid in the form of protons and non-metabolizable acidic anions. The kidneys produce increasing amounts of ammonia to excrete this acid. This process requires the breakdown of the nitrogenous amino acid glutamine, which the body provides by breaking down muscle tissue. Hitherto not examined, we hypothesized that a high dietary acid load (DAL) could alter the serum concentrations of selected amino acids. Using secondary data from a 4-week dietary intervention study conducted in 2017, we examined the associations between various amino acids and DAL in *n* = 42 individuals who either consumed a meat-rich or vegan diet. Results from this secondary data analysis suggested that DAL (as measured by the potential renal acid load and net endogenous acid production) is positively correlated with higher serum concentrations of lysine and 1-methyl-histidine (r = 0.50 and 0.43, respectively) and negatively correlated with glutamine and glycine (r = −0.43 and −0.47, respectively). The inverse association with glycine and glutamine warrants special attention, as both play an important role in many metabolic disorders and the immune system.

## 1. Introduction

There is increasing evidence that even slight changes in the acid–base equilibrium may be detrimental to human health [1,2]. Chronic low-grade metabolic acidosis is now a common phenomenon in the Western world [3] and has been linked to subclinical inflammation as well as cell transformation [4]. Acid-based homeostasis is markedly influenced by dietary factors [5], and the dietary acid load (DAL) has lately received increasing consideration as an important contributor to systemic pH, metabolism, and human acid–base equilibrium [6]. Plant-based foods decrease DAL, whereas meat and cheese products as well as certain grains increase it [7].

While lipids and carbohydrates do not generate unmetabolizable acidity during their complete oxidation [8], proteins contain various amino acids whose catabolism is liable to affect the acid–base equilibrium [9]. The metabolization of protein results in the release of acid in the form of protons; the total amount thereof depends on the amino acid composition [5]. Sulfur-containing amino acids (methionine, homocysteine and cysteine) are major DAL contributors, as their catabolism generates not only protons but also sulfate, which is a non-metabolizable (acidic) anion [5,7,9].
Cysteine → 2H^+^ + SO_4_^2−^ + C₆H₁₂O₆ + CH₄N₂O

Lysine, arginine and histidine are also acidifying because hydrochloric acid is generated upon their hepatic metabolization [5,8].
Arginine^+^ → H^+^ + Cl^−^ + C₆H₁₂O₆ + CH₄N₂O

In comparison, other amino acids are important indirect neutralization agents, as their metabolism involves the consumption of hydrogen ions to become neutral [5,8]. Prominent examples include glutamine (C_5_H_10_N_2_O_3_) and glutamate (C_5_H_9_NO_4_), which may help to neutralize acid via α-ketoglutaric acid (C_5_H_6_O_5_) [10].
C_5_H_10_N_2_O_3_ → C_5_H_9_NO_4_ → C_5_H_6_O_5_^2−^ + 2NH_4_^+^

Although evident from biochemical pathway analyses, animal models and pre-clinical examinations, it remains unclear whether a high DAL does in fact alter serum concentrations of the aforementioned amino acids, which are partly essential and altogether important for human health. Based on the previously discussed pathways and findings from Passey [5], we hypothesized that a high DAL could be correlated with:(I)lower concentrations of glutamine, glutamate and glycine (which are catabolized to buffer the acid load);(II)higher concentrations of lysine, arginine and histidine (which are more abundant in acidic diets).

Such information would be of interest, since serum concentrations of these amino acids have been associated with type 2 diabetes and other cardiovascular risk factors [11,12,13]. To test our hypotheses, we performed a secondary data analysis of a 4-week clinical nutrition intervention study conducted at our department in 2017 [14], which included two diametrically opposed diets: a vegan diet excluding all animal products and a meat-rich diet which included at least 150 g of meat per day. Data from this previously conducted trial was used and re-analyzed to investigate whether measured serum amino acid concentrations correlate with DAL as expected based on the aforementioned biochemical pathways and metabolic processes.

## 2. Materials and Methods

### 2.1. Study Population and Design

The study design and its main features have been described elsewhere in great detail [15,16]. In brief, we performed a monocentric, randomized controlled pilot trial using an isocaloric vegan or meat-rich diet in a parallel group design. The major initial aim of the study was to examine potential associations between dietary intake patterns and systemic inflammation as well as nutrition-related mechanisms involved in the immune system. This study was unique in that two very different but isocaloric diets were compared head-to-head in healthy individuals aged 18 to 60 years. Participants were randomized with the requirement to not alter their total energy intake for 4 weeks.

The vegan diet was defined as a diet excluding all animal products, although there was no strict whole-food plant-based focus. Participants in the meat-rich group were instructed to consume at least 150 g of meat per day, whereby any kind of meat was allowed. More details on both diets may be obtained from Lederer et al. [15,16].

Participants received instruction material and individual education sessions for their assigned dietary patterns at the beginning of the study. All participants were free to choose foods within their assigned diet. Pre-cooked meals were not provided, and participants received no financial remuneration.

### 2.2. Dietary Acid Load Estimations

Dietary acid load was measured based on the work of Remer et al. [17,18], which implies the usage of the potential renal acid load (PRAL) score and the net endogenous acid production (NEAP) score. The PRAL score (hereafter called PRAL_R_) is a nutrient-based estimation score and includes intestinal absorption rates for the following micro-nutrients: potassium, phosphate, magnesium, calcium and protein. Previous studies in healthy individuals revealed a strong correlation between the PRAL score and urinary pH [17].
PRAL (mEq/day) = (0.49 × total protein intake) + (0.037 × phosphorus intake) − (0.021 × potassium intake) − (0.026 × magnesium intake) − (0.013 × calcium intake)

The NEAP_R_ score sums the average intestinal absorption rates of ingested protein and additional minerals (PRAL_R_-score) as well as anthropometry-based estimates for organic acid excretion (OAest).
NEAP_R_ (mEq/d) = PRAL_R_ (mEq/d) + OAest (mEq/d)

We calculated OAest (mEq/d) as follows:Individual body surface area × 41/1.73

The individual body surface area was estimated with the Du Bois and Du Bois formula:Body surface area (m^2^) = (0.007184 × height (cm)^0.725^ × weight (kg)^0.425^)

The NEAP_R_ score estimates the net combination of non-carbonic fixed acids from acids ingested in the diet and produced from endogenous metabolic processes, minus the acids that are neutralized or buffered by non-carbonic dietary and endogenously generated base supplies [7,19].

Finally, we used a third DAL estimation method proposed by Frassetto et al. to estimate net endogenous acid production (NEAP_F_) based on potassium and protein intake [20]:NEAP_F_ (mEq/d) = (54.4 × protein (g/d)/potassium (mEq/d)) − 10.2

### 2.3. Dietary Intake

Dietary intake was assessed using weekly nutritional protocols. The “Freiburger Ernährungsprotokoll” (a validated German food frequency questionnaire) was used to capture food intake [21]. Food intake data was subsequently entered into a special dietetic software (NutriGuide^®^ software (Version 4.7, Nutri-ScienceGmbH, Hausach, Germany) to calculate nutrient intake during the 4th week of the dietary intervention. Participants were requested to complete the provided nutritional protocol on every single day during the last week of the study. Only participants with a complete nutritional protocol (not more than one missing day per week) and with a plausible energy intake (≥800 kcal/day) based on Willett’s criteria [22] were considered eligible for this analysis. Due to the isocaloric intervention and in light of our previous findings (no statistically significant energy intake differences across participants of different groups), no energy adjustments were performed.

### 2.4. Amino Acid Profiles

Blood samples for serum amino acid profiling were drawn on the last day of the study (day 28) in the morning hours (8 a.m. until 10 p.m.). Serum samples were collected and aliquoted in 1.5-mL cryovials. They were stored at −20 °C until transport to the Laboratory of Clinical Biochemistry and Metabolism of the Department of Pediatrics at the University Hospital of Freiburg. Here, all examination methods were established and validated.

Serum amino acid profiles were measured after precipitating proteins by adding 50 µL of 10% aqueous sulfosalicylic acid to 200 µL of serum. All samples were centrifuged for 8 min at 3600× *g* and the supernatant was analyzed with a Biochrom 30 amino analyzer (Biochrom Ltd., Cambridge, UK), using ion exchange chromatography and post-column ninhydrine derivatization. We performed single measurements for each sample. Quality controls from ClinChek were included in each assay (ClinCheck, Level I and Level II, plasma control, lyophilized for amino acids, product Nrs. 10280 and 10281, Recipe^®^, GmbH, Munich, Germany).

### 2.5. Statistical Analysis and Analytical Considerations

The present study is a post-hoc analysis of a randomized-controlled trial, with a slightly different sample size and analysis structure. The initial study was planned for three different immunological main outcome parameters considering a statistical power of 80% and a hypothesized large effect size. An a priori sample size calculation revealed that at least 48 participants (24 for each diet) would be required to detect a statistical difference of *p* < 0.05 between the groups [14]. The final sample included *n* = 53 participants, including *n* = 27 individuals on a meat-rich diet and *n* = 26 individuals on a vegan diet. Unlike our previous publications, however, we did not compare the two groups in this analysis head-to-head but intentionally used the whole sample to perform correlation analyses with the amino acid profiles in a cross-sectional manner at week 4 of the dietary intervention. This was deemed necessary to increase the sample size for analysis and to compile a dataset with large contrasts in individual key measurements of interest (in particular in the DAL scores).

As shown previously, DAL is substantially lower with a vegan diet as compared to a meat-rich diet [16,23]. Such contrasts were non-existent at baseline, where all participants consumed the same diversified diet. However, in light of the limited study sample size, they were deemed necessary to potentially detect the hypothesized correlations. For this analysis, only participants with a full data set were considered. This implied the following criteria: available serum data for all investigated amino acids and available nutrient intake data at week 4 (and thus available DAL scores), as well as no missing anthropometric data.

We used STATA 14 statistical software (StataCorp. 2015. Stata Statistical Software: Release 14. College Station, TX, USA: StataCorp LP) for the analysis. Histograms and the Shapiro–Wilk test were used to check for normality of continuous variables. Normally distributed continuous variables were described with their mean and standard deviation. For non-normally distributed variables, we presented medians and their corresponding interquartile ranges. Afterward, we ran multiple Pearson’s product moment correlations and Spearman correlations to assess the relationship between DAL scores and amino acids in all participants.

### 2.6. Ethical Approval

The ethical committee of the University Medical Center of Freiburg, Germany (EK) Freiburg 38/17) approved the trial, which was registered at the German Clinical Trial register before onset (DRKS00011963).

## 3. Results

After excluding participants with missing data, a total sample of *n* = 42 participants remained eligible for the final analysis (Figure 1). Table 1 summarizes the sample characteristics, whereas Table 2 shows nutrient intake data and the resulting DAL scores of all participants.

Higher PRAL_R_ scores were strongly correlated with higher NEAP_R_ scores (r = 0.986, *p* < 0.001), as visualized in Figure 2. As reported earlier, vegans yielded substantially lower DAL scores as compared to those individuals consuming a meat-rich diet [16]. Those individuals assigned to a vegan diet consumed an alkaline diet (PRAL_R_: −30.48 ± 3.30 mEq/d), whereas those on a meat-rich diet consumed an acidifying diet (PRAL_R_: 17.10 ± 2.32 mEq/d). NEAP_R_ scores were 12.28 ± 3.31 mEq/d in the vegan group and 61.27 ± 2.86 mEq/d in the meat-rich group. Median NEAP_F_ scores were more than twice as high in the meat-rich group (55.81 (15.38) mEq/d) as opposed to the vegan group (25.26 (5.14) mEq/d).

Serum amino acid concentrations for the entire sample are displayed in Table 3.

### 3.1. PRAL_R_ and Its Association with Amino Acid Serum Concentrations

The associations between PRAL_R_ and the serum concentrations of the sulfur-containing amino acids (cysteine, homocysteine and methionine) are shown in Figure 3. Figure 4 and Figure 5 show the relationship between PRAL_R_ and the basic (positively charged) amino acids lysine, arginine and histidine (Figure 4) as well as with glutamate, glutamine and glycine (Figure 5). Significant associations were found for the four following amino acids: 1-methyl-histidine, lysine, glutamine and glycine (*p* < 0.05).

### 3.2. NEAP_R_ and Its Association with Amino Acid Serum Concentrations

In a similar style, Figure 6, Figure 7 and Figure 8 display associations between NEAP_R_ and the sulfur-containing amino acids (Figure 6); the positively charged amino acids lysine, arginine and histidine (Figure 7); and finally with glutamate, glutamine and glycine (Figure 8). Comparably to the PRAL_R_ score, we found significant associations for 1-methyl-histidine, lysine, glutamine and glycine.

### 3.3. NEAP_F_ and Its Association with Amino Acid Serum Concentrations

Significant associations between the NEAP_F_ score and serum concentrations of amino acids were found for glycine (r = −0.395, *p* = 0.01), glutamine (r = −0.353, *p* = 0.022), 1-methyl-histidine (r = 0.486, *p* = 0.001) and lysine (r = 0.507, *p* = 0.001). We observed no significant associations with the remaining amino acids.

## 4. Discussion

A high DAL has been associated with various metabolic alterations that may predispose individuals to an increased risk of non-communicable diseases [24]. In light of the previously discussed compensatory mechanisms of the human body to counteract protons and non-metabolizable anions subsequent to high intake of acidic foods, we hypothesized that a high DAL could be associated with higher serum concentrations of lysine, arginine and histidine as well as with lower concentrations of glutamine, glutamate and glycine. These hypotheses were put to the test using secondary data from a randomized controlled trial that included a vegan and a meat-rich diet—two diets that yield distinct amino acid profiles and thus allow for such a comparison [15].

Our results largely confirmed our hypothesis with regard to lysine, glutamine, glycine and 1-methyl-histidine. Notably, no significant associations were found for arginine and glutamate. These results are noteworthy and have potential implications for human health.

Numerous studies have confirmed that vegans and lacto-ovo-vegetarians generally consume lower amounts of lysine, glutamine, glycine and histidine [25,26]. Table 4 shows exemplary intake data from vegans in comparison to omnivores from two well-described European cohorts: the EPIC–Oxford cohort including 392 men aged 30–49 years analyzed by Schmidt et al. [25] as well as the “Risks and Benefits of a Vegan Diet” (RBVD) German cohort aged 30–60 years, which was previously described by Dietrich et al. [26].

As expected, omnivores had a higher dietary intake of histidine and lysine [25]. Despite their lower dietary intake (Table 4), vegans yielded higher serum concentrations of glutamine and glycine in both of the aforementioned studies [25,26]. Questions arise as to whether DAL could play a role in this reciprocal relationship.

Amino acids are important indirect neutralization agents since their metabolism allows for the consumption of hydrogen ions to become neutral [5]. Glutamine is a prominent example, which may help to neutralize acid via α-ketoglutaric acid (C_5_H_6_O_5_) [10].
C_5_H_10_N_2_O_3_ → C_5_H_9_NO_4_ → C_5_H_6_O_5_^2−^ + 2NH_4_^+^

Although cross-sectional, it is not inconceivable that this pathway played an important role in our study. Participants assigned to a meat-rich diet were exposed to a high DAL, with a subsequent release of acid in the form of protons. One main pathway to excrete this acid is via the kidneys and the increased formation of ammonia [13]. The ammonia production in the kidney’s tubular cells comes primarily from the breakdown of the nitrogenous amino acid glutamine, which the body provides by breaking down muscle tissue [3]. Useful and beneficial to buffer protons in the short term, this process has deleterious long-term consequences if not offset by muscle training or a higher intake of alkalizing foods. Chronic exposure to a high DAL can lead to frailty and impaired musculoskeletal health, particularly in older adults [27,28].

We provide a potential mechanistic explanation for this phenomenon using data from our 2017 nutritional intervention trial, and shed new light on the interrelationship among amino acid intake, serum concentrations and the potential role of DAL.

The lower serum concentrations of glycine and glutamine in participants on a meat-rich diet may also play an important role with regard to several other health outcomes. Erikkson et al. reported an indirect association of serum glycine with femoral neck bone mineral density and cortical bone strength, and direct associations with fracture risk in men [29]. Chen et al. suggested that high glutamine concentrations were associated with a decreased risk of incident type 2 diabetes in the Hitachi Health Study [11]. Both studies thus demonstrated that glycine and glutamine may play important roles beyond muscle health. Glutamine in particular is considered as a “fuel for the immune system”, with low blood concentrations impairing immune cell function, resulting in poor clinical outcomes and increased risk of mortality [30,31].

While a more elaborate review of glutamine- and glycine-related health outcomes is beyond the scope of this article, both studies highlight that serum concentrations of the aforementioned amino acids are relevant to human health in many different ways.

A high DAL is inversely associated with the serum concentrations of these amino acids and could thus contribute to various unfavorable health outcomes. Plant-based diets, on the other hand, fare much better with regard to DAL as they minimize the intake of acidifying processed meats and cheese products [32,33,34]. For our study, DAL scores in vegans vs. meat-eaters have been previously reported [16]. Our findings thus have a translational clinical value beyond the discussed associations with serum amino acids and should be taken into account when discussing DAL-associated mechanisms in health and disease. Although of a pilot character, our data indirectly suggest that lowering DAL could be an important strategy to prevent muscle tissue breakdown, which the body performs to provide the nitrogenous amino acid glutamine in order to buffer protons.

The present secondary data analysis has several strengths and weaknesses to consider. As for the strengths, we present a unique dataset and a randomized controlled intervention setting in which the aforementioned hypotheses were tested. The usage of three different DAL scores and its exploratory character make this analysis particularly compelling. Weaknesses include the cross-sectional character of the analysis, which does not permit causal interferences. Moreover, we acknowledge a limited sample size and the typical limitations of clinical nutrition intervention studies: reporting bias, recall bias and dietary adherence problems as indicated by incomplete protocols. Due to the limited sample size, results should be interpreted with caution, and larger confirmative studies should be conducted in the near future. The uneven sex distribution of this cohort, with only 15 males, did not allow for the important investigation of sex-specific differences, as has been possible, at least for plasma amino acid concentrations, in large-size cross-sectional studies [25]. Non-dietary modifiers of amino acid profiles, such as the level of physical exercise, were not investigated in this study. It has been reported that either very high-intensity short-term exercise or prolonged exercise are required to induce a significant decrease in the plasma amino acid pool, with the most susceptible amino acid class being the branched-chain amino acids due to the activation of fatty acid oxidation [35]. Since none of the participants in the study were high-performance athletes, and per our study protocol all participants were instructed to maintain their level of physical activity during the trial, we speculate little to no effect of physical exercise on the absolute plasma amino acid concentrations was present in this study. The fact that the trial was conducted in young and healthy individuals is another non-neglectable aspect. Whether our results may also apply to non-healthy populations remains subject to further investigations. Finally, it is noteworthy that this trial included an isocaloric vegan diet in which participants were instructed to not lose weight. This implies that several participants assigned to the vegan group occasionally consumed acidifying grain-based snacks, processed wheat products and sweets in order to reach the daily energy intake goal of approximately 2000 kcal/d. Our results may thus underestimate the “true” alkalizing effect of a more plant-based vegan diet abundant in fruits, legumes and vegetables.

## 5. Conclusions

Our results suggest moderate and significant associations between DAL and the serum concentration of various amino acids. The negative correlation between a high DAL and glycine and glutamine warrants special attention, as both amino acids play an important role in many metabolic disorders. Despite the limitations of a cross-sectional analysis, our data support the role of DAL in altering serum concentrations of amino acids, calling for larger studies with metabolomic endpoints in the near future.

## Figures and Tables

**Figure 1 nutrients-15-02942-f001:**
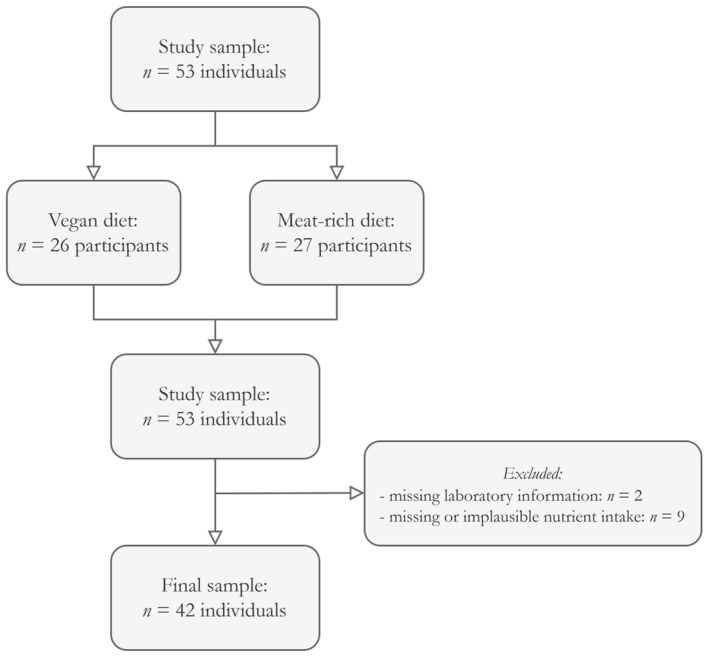
Participant inclusion flowchart. We only considered participants with a complete dataset eligible for this secondary data analysis.

**Figure 2 nutrients-15-02942-f002:**
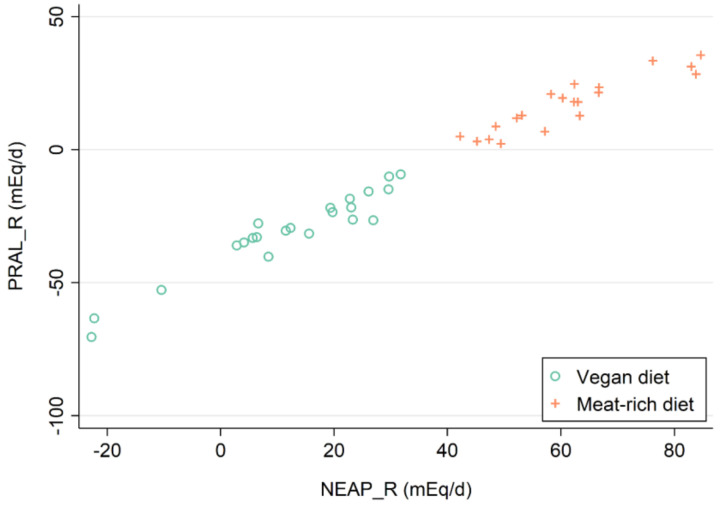
DAL score distribution in the study sample by dietary group. Participants who were assigned to a vegan diet are marked in green, whereas individuals on a meat-rich diet are marked in orange.

**Figure 3 nutrients-15-02942-f003:**
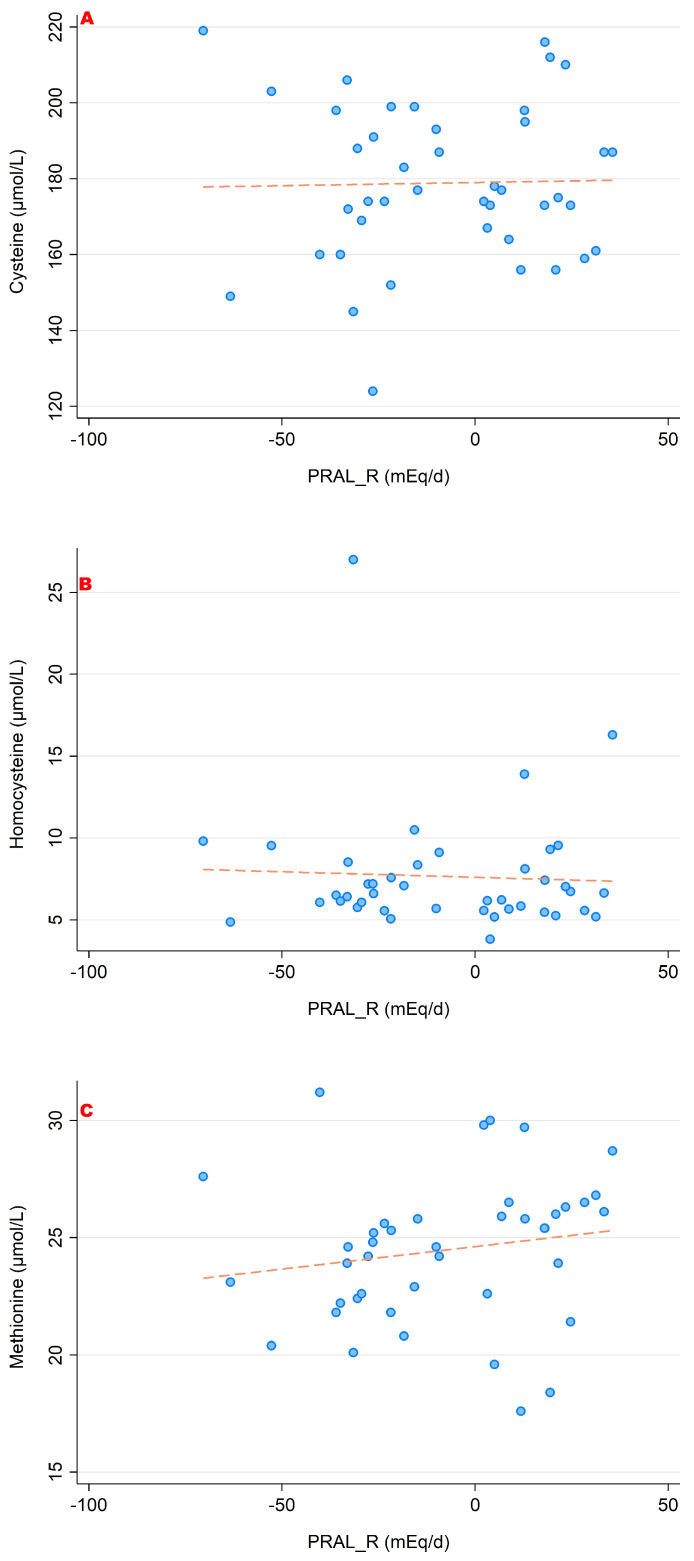
Associations between PRAL_R_ and the sulfur-containing amino acids ((**A**) cysteine; (**B**) homocysteine; (**C**) methionine). No significant correlations were found with regard to all three amino acids. All amino acid serum concentrations are shown in μmol/L.

**Figure 4 nutrients-15-02942-f004:**
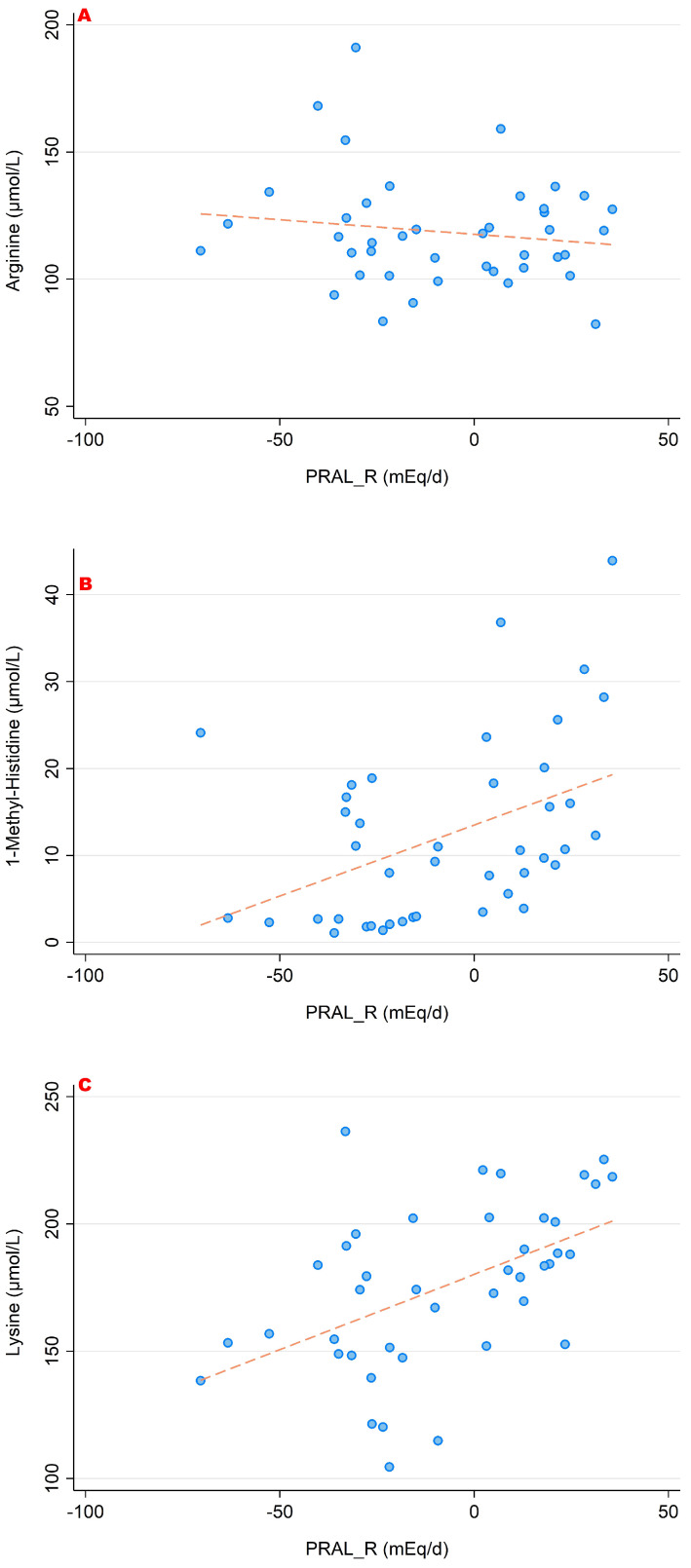
Associations between PRAL_R_ and the amino acids arginine (**A**), histidine (**B**) and lysine (**C**). Significant moderate correlations were found for 1-methyl-histidine (r = 0.432, *p* = 0.004) and lysine (r = 0.502, *p* = 0.001). No significant correlations were found with regard to arginine. All amino acid serum concentrations are shown in μmol/L.

**Figure 5 nutrients-15-02942-f005:**
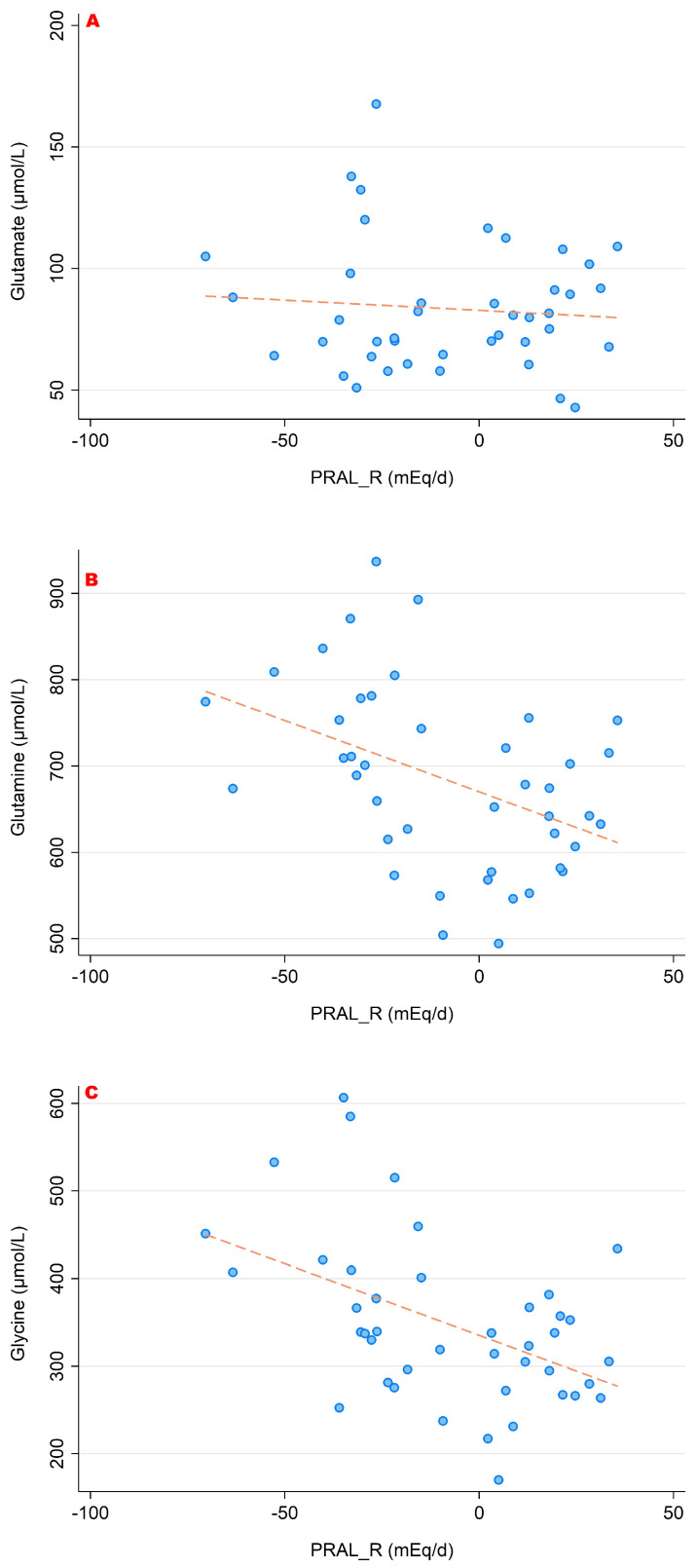
Associations between PRAL_R_ and the amino acids glutamate (**A**), glutamine (**B**) and glycine (**C**). Significant moderate inverse correlations were found for glutamine (r = −0.432, *p* = 0.004) and glycine (r = −0.472, *p* = 0.002). We observed no significant correlations with regard to glutamate. All amino acid serum concentrations are shown in μmol/L.

**Figure 6 nutrients-15-02942-f006:**
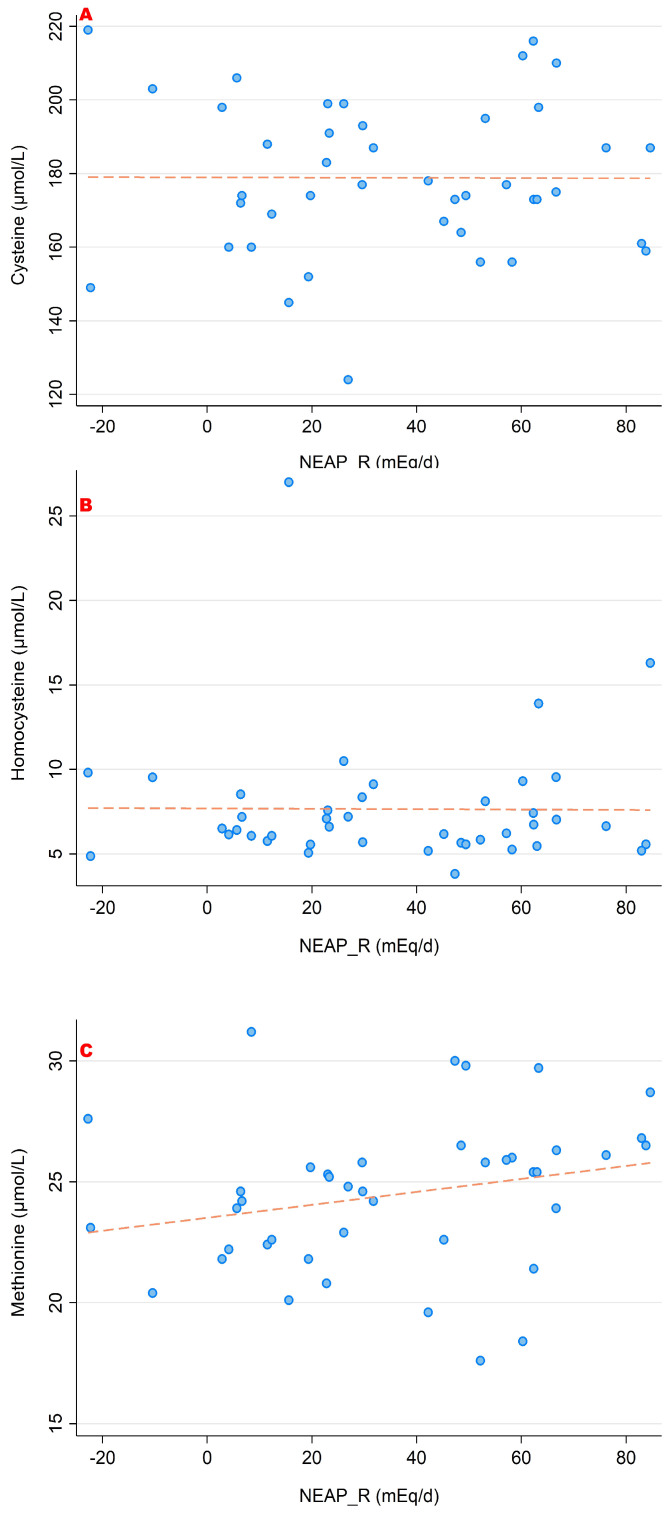
Associations between NEAP_R_ and the sulfur-containing amino acids ((**A**) cysteine; (**B**) homocysteine; (**C**) methionine). No significant correlations were found with regard to all 3 amino acids. All amino acid serum concentrations are shown in μmol/L.

**Figure 7 nutrients-15-02942-f007:**
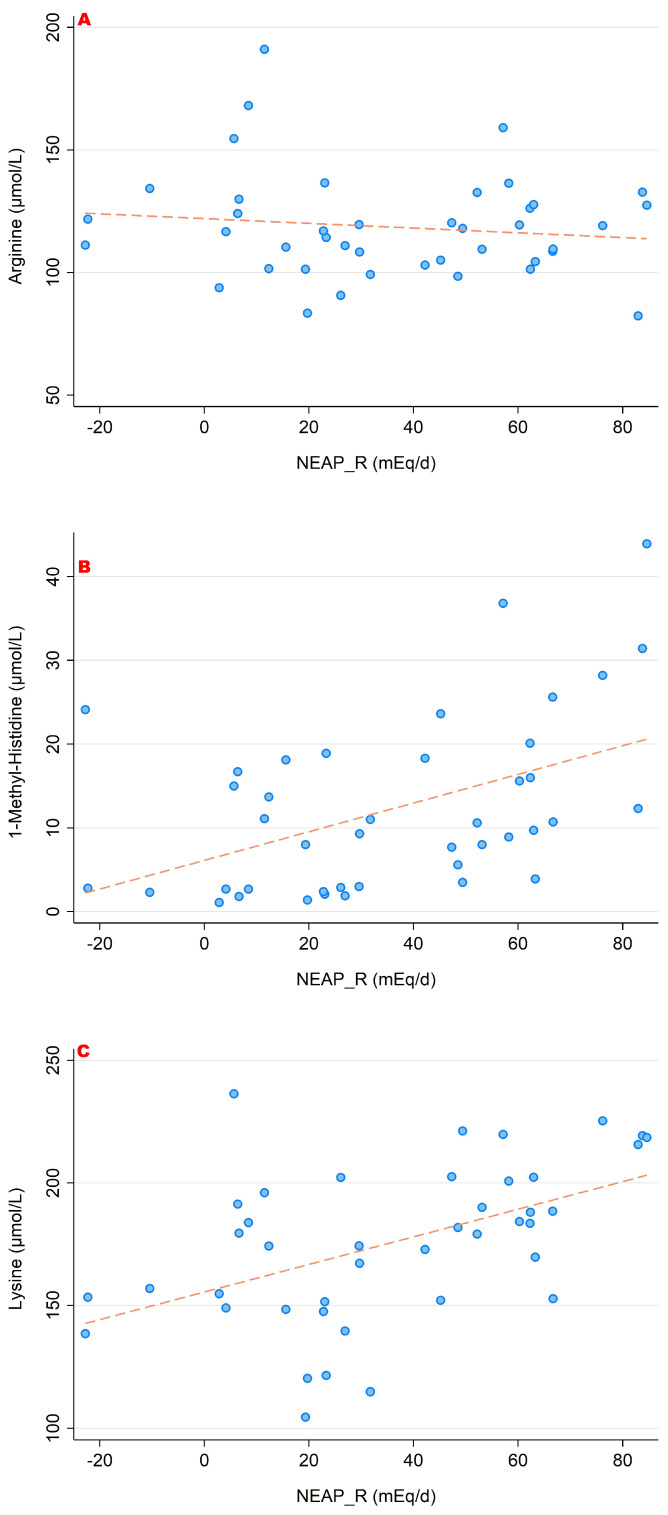
Associations between NEAP_R_ and the amino acids arginine (**A**), histidine (**B**) and lysine (**C**). Significant moderate correlations were found for 1-methyl-histidine (r = 0.471, *p* = 0.002) and lysine (r = 0.499, *p* = 0.001). No significant correlations were found with regard to arginine. All amino acid serum concentrations are shown in μmol/L.

**Figure 8 nutrients-15-02942-f008:**
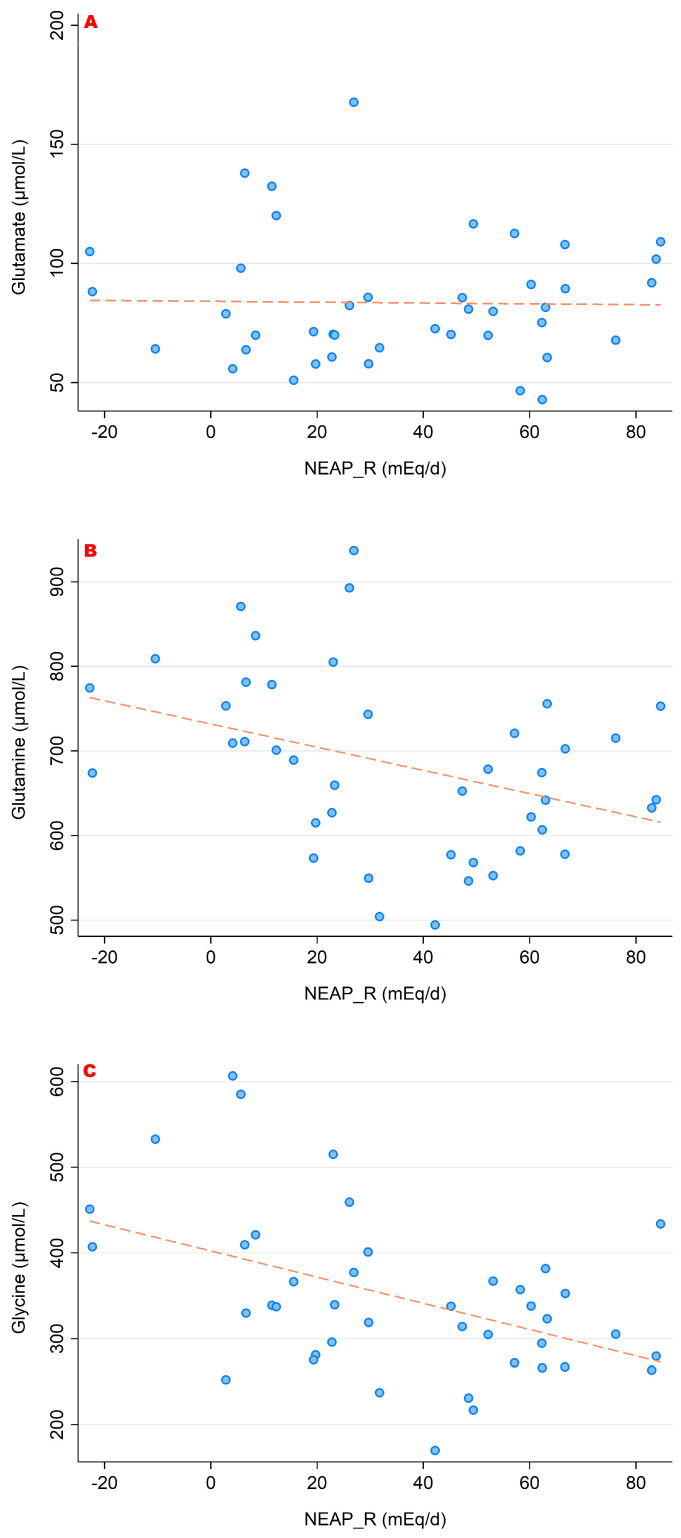
Associations between NEAP_R_ and the amino acids glutamate (**A**), glutamine (**B**) and glycine (**C**). Significant moderate inverse correlations were found for glutamine (r = −0.373, *p* = 0.015) and glycine (r = −0.458, *p* = 0.002). We observed no significant correlations with regard to glutamate. All amino acid serum concentrations are shown in μmol/L.

**Table 1 nutrients-15-02942-t001:** Sample characteristics.

Variable	Result
Age (years)	28.5 (12)
Gender	
Males	*n* = 15 (35.71%)
Females	*n* = 27 (64.29%)
Height (cm)	173.24 ± 9.78
Weight (kg)	66.15 (15.2)
Body Mass Index (kg/m^2^)	22.77 (3.28)

Total study sample: *n* = 42. Normally distributed continuous variables are shown with their mean ± standard deviation; non-normally distributed continuous variables are shown with their median (interquartile range). Categorical variables are displayed in the following format: *n* (percentage). *n* = number of observations.

**Table 2 nutrients-15-02942-t002:** Nutrient intake data and DAL scores.

Nutrient and Energy Intake	
Total energy intake (kcal/d)	2318 (850)
Magnesium intake (mg/d)	382.79 (246.04)
Potassium intake (mg/d)	3419.17 (1405.29)
Calcium intake (mg/d)	691.38 (469.08)
Phosphorus intake (mg/d)	1347.75 (618.05)
Protein intake (g/d)	86.93 (41.98)
**DAL scores**	
PRAL_R_ (mEq/d)	−7.82 ± 27.41
NEAP_R_ (mEq/d)	35.61 ± 28.51
NEAP_F_ (mEq/d)	37.67 (30.29)

Total study sample: *n* = 42. Normally distributed continuous variables are shown with their mean ± standard deviation; non-normally distributed continuous variables are shown with their median (interquartile range).

**Table 3 nutrients-15-02942-t003:** Serum amino acid concentrations.

Amino Acid	
Arginine (μmol/L)	116.85 (23.2)
Cysteine (μmol/L)	178.88 ± 20.82
Glutamate (μmol/L)	79.4 (33.4)
Glutamine (μmol/L)	683.07 ± 105.01
Glycine (μmol/L)	337.45 (121.7)
Histidine (μmol/L)	10.15 (15.2)
Homocysteine (μmol/L)	6.56 (2.7)
Lysine (μmol/L)	175.55 ± 32.16
Methionine (μmol/L)	24.46 ± 3.10

Total study sample: *n* = 42. Normally distributed continuous variables are shown with their mean ± standard deviation; non-normally distributed continuous variables are shown with their median (interquartile range). All amino acid serum concentrations are displayed in μmol/L.

**Table 4 nutrients-15-02942-t004:** Dietary amino acid intake in vegans: an overview of two well-described European cohorts.

Amino Acid	EPIC–Oxford Cohort		RBVD Cohort	
	Vegans	Omnivores	Vegans	Omnivores
Lysine	2.82 (2.69, 2.95) ^a^	5.01 (4.78, 5.24) ^a^	41.0 (32.2–67.1) ^c,b^	78.9 (63.0–97.5) ^c,b^
Glutamate/Glutamine	14.06 (13.61, 14.52) ^a^	16.10 (15.59, 16.63) ^a^	206.7 (161.4–269.5) ^c,b^	250.1 (225.3–286.4) ^c,b^
Glycine	2.61 (2.50, 2.71) ^a^	3.12 (3.00, 3.25) ^a^	41.1 (32.2–57.0) ^c,b^	45.6 (40.3–61.3) ^c,b^
Histidine	1.52 (1.46, 1.57) ^a^	2.12 (2.04, 2.20) ^a^	20.0 (17.4–31.1) ^c,b^	32.0 (27.5–39.0) ^c,b^

^a^ = geometric mean intake (95% confidence interval), in g/d; ^b^ = data is reported as median (IQR); ^c^ = in mg/d per kg body weight. The vegan group comprised *n* = 98 individuals in the EPIC–Oxford analysis and *n* = 36 individuals in the RBVD analysis. The omnivorous group comprised *n* = 98 individuals in the EPIC–Oxford analysis and *n* = 36 individuals in the RBVD analysis.

## Data Availability

The data presented in this study will be made available upon reasonable request from the corresponding author.

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
