# Peer review of "Dietary Acid Load Correlates with Serum Amino Acid Concentrations after a Four-Week Intervention with Vegan vs. Meat-Rich Diets: A Secondary Data Analysis"

_nutrients, 2023, doi:10.3390/nu15132942_

Round 1

Reviewer 1 Report

Overall, a well-written easy to follow paper.   I have a very few suggestions. 

First, the section with Figures 3 through 8 is not formatted correctly.  The third graph on each of these sets of figures is only partially visible. Please address this.  

Second, in Table 4 you have written Glutamine/Glutamine instead of Glutamate/Glutamine.  Please address this.

Other than that, I thought it was an interesting and useful paper. 

Author Response

Dear Reviewer #1,

We would like to thank you and the other reviewers very much for careful and thorough reading of this manuscript and for the thoughtful comments and constructive suggestions, which helped us to improve the quality of this article. We made all the requested revisions to our original manuscript based on all the comments we received. All changes have been clearly marked in yellow and blue color. Please find our specific point-by-point response attached.

Sincerely,

The authors

Reviewer 2 Report

It is a significant study where authors investigated potential association of dietary acid load with serum amino acids, which could provide some implication for nutrition and related health issues. Some technical issues should be addressed if a revision is invited.

1.      It is unclear when to collect dietary information. It is collected at 4th week of intervention? Just once? Or multiple times?

2.      Also it is not clear about time when to measure serum amino acid. If possible, authors are suggested to state this issue more clearly.

3.      A key issue is how to consider two kinds of intervention group? Is that possible to observe the correlation between DAL and serum amino acid by intervention group.

4.      The association of interest is modified by some factors such as age, sex or physical activities?

5.      Whether the intake of micronutrients used for DAL is energy adjusted?

6.      Due to smaller sample size, the results should be interpreted with caution.

Author Response

Dear Reviewer #2,

We would like to thank you and the other reviewers very much for careful and thorough reading of this manuscript and for the thoughtful comments and constructive suggestions, which helped us to improve the quality of this article. We made all the requested revisions to our original manuscript based on all the comments we received. All changes have been clearly marked in yellow and blue color. Please find our specific point-by-point response attached.

Sincerely,

The authors

Reviewer 3 Report

The manuscript is interesting, the methodology is sufficient and updated. The experimental design is adequate. However, in the results section, I had a major problem. The results shown in figures 3, 4, 5, 6, 7 and 8. They are incomplete. This problem did not allow me to continue with the revision of the manuscript. Please correct this problem. In addition, in each figure include a letter that identifies each graph. Example: Fig.3A, 3B and 3C.

Only by correcting this problem will it be possible to perform an optimal evaluation of the manuscript.

The manuscript is well written, but I suggest a minor edition of the English.

Author Response

Dear Reviewer #3,

We would like to thank you and the other reviewers very much for careful and thorough reading of this manuscript and for the thoughtful comments and constructive suggestions, which helped us to improve the quality of this article. We made all the requested revisions to our original manuscript based on all the comments we received. All changes have been clearly marked in yellow and blue color. Please find our specific point-by-point response attached.

Sincerely,

The authors

Round 2

Reviewer 2 Report

Thank authors for their reply and revision. I have not any other concerns.

Author Response

Dear Reviewer,

We would like to thank you again very much for careful and thorough reading of this manuscript and for your positive feedback.

Sincerely,

The authors

Reviewer 3 Report

The manuscript presents very interesting results. However, it is important that authors make improvements to the manuscript.

I. Major comments:

1. Improve the title of the manuscript. The title does not correspond to the nutritional intervention carried out. For example, it is important to include the type of diet.

2. The introduction is insufficient, especially due to the lack of background information on the diets of the subjects in the study. The introduction must present antecedents that justify the realization of the study.

3. Methodology ....randomized-controlled pilot trial using an isocaloric vegan or meat-rich diet in a parallel group design...., this aspect is relevant in the study, and the authors do not refer to this point in the title, introduction and discussion. This is a major weakness in the manuscript.

4. Table 2. Why do the authors not show the intake of other nutrients (proteins, lipids, etc.)?

5. It is necessary to include blood parameters, especially albumin.

6. The study was carried out in young and healthy people (normal-weight). However, the authors do not discuss relevant aspects focused on people with more age and body weight.

7. What clinical application does this study have? This point is not disputed by the authors.

II. Minor comments:
1. Improve the wording of the objective of the study.
2. In the introduction it is not necessary to include the metabolization formulas. Example: metabolism of arginine.
3. Figure 3. Include a letter to better understand the figures. Example. A, B, and C. Include this description in the legend to Figure 3. Also, make these changes to the other figures.
4. I suggest grouping figures and reducing the number of figures (figures 3 to 8).
5. Improve the resolution of the figures.  

Overall Recommendation: Reconsider after major revision (control missing in some experiments)

The manuscript is well written, but some editorial errors need to be corrected.

Author Response

Dear Reviewer #3,

We would like to thank you again very much for careful and thorough reading of this manuscript and for the thoughtful comments and constructive suggestions, which help us to improve the quality of this article. All requested changes have been clearly marked in yellow and blue color. We appreciate your input, your advice and the fast peer review. Please find our point-by-point response attached. Thank you!

Sincerely,

The authors
